# Cochlear implant positioning and fixation using 3D-printed patient specific surgical guides; a cadaveric study

**Laura M. Markodimitraki**[1,2]*, **Timen C. ten Harkel**[3], **Ronald L. A. W. Bleys**[4], **Inge Stegeman**[1,2,5,6], **Hans G. X. M. Thomeer**[1,2]

**1** Department of Otorhinolaryngology and Head & Neck Surgery, University Medical Center Utrecht, Utrecht, the Netherlands, **2** UMC Utrecht Brain Center, Utrecht University, Utrecht, the Netherlands, **3** Department of Oral and Maxillofacial Surgery, University Medical Center Utrecht, Utrecht, the Netherlands, **4** Department of Anatomy, University Medical Center Utrecht, Utrecht, the Netherlands, **5** Department of Ophthalmology, University Medical Center Utrecht, Utrecht, the Netherlands, **6** Epidemiology and Data Science, Amsterdam University Medical Centers, University of Amsterdam, Amsterdam, the Netherlands

\* l.m.markodimitraki-3@umcutrecht.nl

## Abstract

### Hypothesis

To develop and validate the optimal design and evaluate accuracy of individualized 3D-printed surgical guides for cochlear implantation.

### Background

Positioning and fixation of the cochlear implant (CI) are commonly performed free hand. Applications of 3-dimensional (3D) technology now allow us to make patient specific, bone supported surgical guides, to aid CI surgeons with precise placement and drilling out the bony well which accommodates the receiver/stimulator device of the CI.

### Methods

Cone beam CT (CBCT) scans were acquired from temporal bones in 9 cadaveric heads (18 ears), followed by virtual planning of the CI position. Surgical, bone-supported drilling guides were designed to conduct a minimally invasive procedure and were 3D-printed. Fixation screws were used to keep the guide in place in predetermined bone areas. Specimens were implanted with 3 different CI models. After implantation, CBCT scans of the implanted specimens were performed. Accuracy of CI placement was assessed by comparing the 3D models of the planned and implanted CI's by calculating the translational and rotational deviations.

### Results

Median translational deviations of placement in the X- and Y-axis were within the predetermined clinically relevant deviation range (< 3 mm per axis); median translational deviation in

**Data Availability Statement:** All relevant data are within the manuscript and its Supporting information files. Some data cannot be shared

publicly because of potentially identifying or sensitive patient information. This applies for the CBCT scans used as well as the 3D models made based on these scans. Data are available from the datamanager of the Otorhinolaryngology department of the University Medical Center Utrecht (DHS-datamanagement@umcutrecht.nl) for researchers who meet the citeria for access to confidential data.

**Funding:** Oticon Medical directly funds the PhD research project of LM, via the University Medical Center Utrecht. The funders had no role in study design, data collection and analysis, decision to publish, or preparation of the manuscript.

**Competing interests:** The authors have declared that no competing interests exist.

the Z-axis was 3.41 mm. Median rotational deviations of placement for X-, Y- and Z-rotation were 5.50˚, 4.58˚ and 3.71˚, respectively.

## Conclusion

This study resulted in the first 3D-printed, patient- and CI- model specific surgical guide for positioning during cochlear implantation. The next step for the development and evaluation of this surgical guide will be to evaluate the method in clinical practice.

## Introduction

Cochlear implantation has been an accepted treatment for patients with severe-to-profound sensorineural hearing loss for several decades [1]. Nowadays, it is regarded as a safe procedure with low complication rates, and surgical techniques are continuously improving to achieve better audiological results [2]. Placement and fixation of the cochlear implant (CI) is an underestimated step during the cochlear implantation procedure. The internal part of the cochlear implant, also known as the receiver/stimulator (R/S) device, is designed to reside in close proximity to the pinna, without any interference with the external processor. During cochlear implantation the CI surgeon positions the R/S device under the temporalis muscle by either drilling out a part of the skull cortex (a bony well) with or without suture retaining holes, or by creating a subperiosteal pocket which holds the device in place. CI manufacturers provide information about the optimal distance from the pinna and the angle relative to the ear canal/pinna. Templates are provided by the manufacturers to draw the outlines of the external and internal parts on the surgical drapes to aid in positioning the implant. However, these templates provide an estimate at best of where the implant will reside [3]. The drawings on the surgical drapes are often arbitrary, imprecise and during the operative procedure it is difficult to match the external drawing to the skull surface. Some surgeons additionally apply a percutaneous marker (small diamond burr or methylene blue stain) through the skin on the bone, thereby locating more exactly the position of the definitive implant position on the temporal cortex during surgery [4, 5]. In case of bilateral implantation, achieving symmetrical placement is challenging as well. Interindividual variability of cortical thickness of the temporal bone between patients, can also be a factor of influence when drilling out a bony well [6]. We believe some of these issues can be solved by using patient-specific, bone-supported, surgical guides.

Intraoperative guides are templates used in a variety of ways for tissue reconstruction, by assisting cutting or drilling. In health care, and surgery specifically, the concept of patient-specific surgical guides is far from new, and it is being applied in everyday medical practice [7]. In the field of otology, 3D-printed guides have already successfully been used for hearing implant surgery [8]. Until now, R/S device placement and drilling is usually performed free hand. The goal of the surgical guide is to aid the CI surgeons with precise placement and drilling procedure of the bony well, which accommodates the R/S device. This study aims to develop and validate a patient specific, bone supported surgical guide.

## Materials and methods

### Specimens

For this feasibility study, we used fresh frozen human cadaveric heads that were obtained through the Human Body Donation program of the University of Utrecht (https://www.

umcutrecht.nl/nl/meedoen-aan-wetenschappelijk-onderzoek). From these persons written informed consent was obtained during life that allowed the use of their entire bodies for educational and research purposes. The possibility for body donation is part of the Dutch law on dead bodies. As no living human subjects were involved, this work was exempt from review by the Institutional Review Board of the UMC Utrecht. The specimens had to have an intact temporal and parietal bone and retroauricular skin. A power analysis was conducted to calculate sample size. We estimated a translational difference of 3.0 mm to be clinically relevant, based on expert opinion, with a standard deviation of 2.0 mm. With an alpha of 0.05 and a power of 85%, we needed to include 17 ears. Rotational deviations above 5˚ in the sagittal plane were deemed clinically relevant.

## Planning and guide design

Specimens underwent Cone Beam CT scans (VGi evo, NewTom, Cefla C.S., Italy) with a 24 x 19 cm field of view (FoV), and 0.3 mm slice thickness. Images were stored in DICOM format. Using the segmentation feature in iPlan (Brainlab, Munich, Germany), the skull was segmented and reconstructed into a 3D model. This 3D model was then imported into 3-matic version 14.0 medical design software (Materialise). The CI's used for this study were Cochlear CI512, Oticon Neuro Zti and MedEl SONATA TI(100). The CI's from Cochlear and MedEl were used models, acquired after revision or explantation surgery due to device failure or patient dissatisfaction with speech recognition results. The CI from Oticon Medical was provided by the manufacturer for research purposes. Volume data of the CI's were acquired by scanning the implants using a 3shape laboratory scanner (3shape, Copenhagen, Denmark). The data was reconstructed into 3D models.

The planning of the implantation was conducted by the following steps. First two virtual planes were created on the 3D model of the skull, namely the Frankfurt Horizontal plane that connects the inferior margins of the orbits and the superior margin of the external auditory canal (EAC), and a 45˚ plane relative to the Frankfurter Horizontal plane, originating from the EAC (Fig 1a). Next, the CI was aligned to the 45˚ plane with a distance of 2.5 cm from the EAC. During the positioning of the CI the curvature of the skull was taken into account. The position for the Cochlear and MedEl models was determined so that the anterior part of the implant (receiver/stimulator) would be embedded whilst allowing the posterior part (magnet with coil) to rest on the skull. The Oticon implant was embedded in the skull in its entirety. In order to achieve symmetrical placement, the 3D model of the cochlear implant was duplicated and mirrored to the contralateral side over the sagittal plane as defined by the Frankfurt Horizontal plane. With the implants in place, the drilling guides were designed. The skull surface of the mastoid bone and the supramastoid crest were used as contact areas and were defined (Fig 1b and 1c). After each implantation the surgical guide was reviewed based on the feasibility and the deviation results. The surface contact area was extended or reduced accordingly to optimize the design. Screw holes were created to stabilize the guide on the area of the mastoid bone. All guides were produced using a medical certified photopolymer resin (Model 2.0, Next-Den, Soesterberg, The Netherlands) using selective laser sintering 3D printing.

## Surgical workflow

Implantations were carried out by a clinical research physician (LM) who had undergone surgical training prior to start of the study. One implantation was carried out by a senior CI surgeon (HT). Fixation of the CI's using the drilling guides was carried out as follows. A retroauricular Lazy-S incision of approximately 8–9 cm was made. The bony surface was exposed to fit the designated location on the temporal bone. The periosteum was elevated to

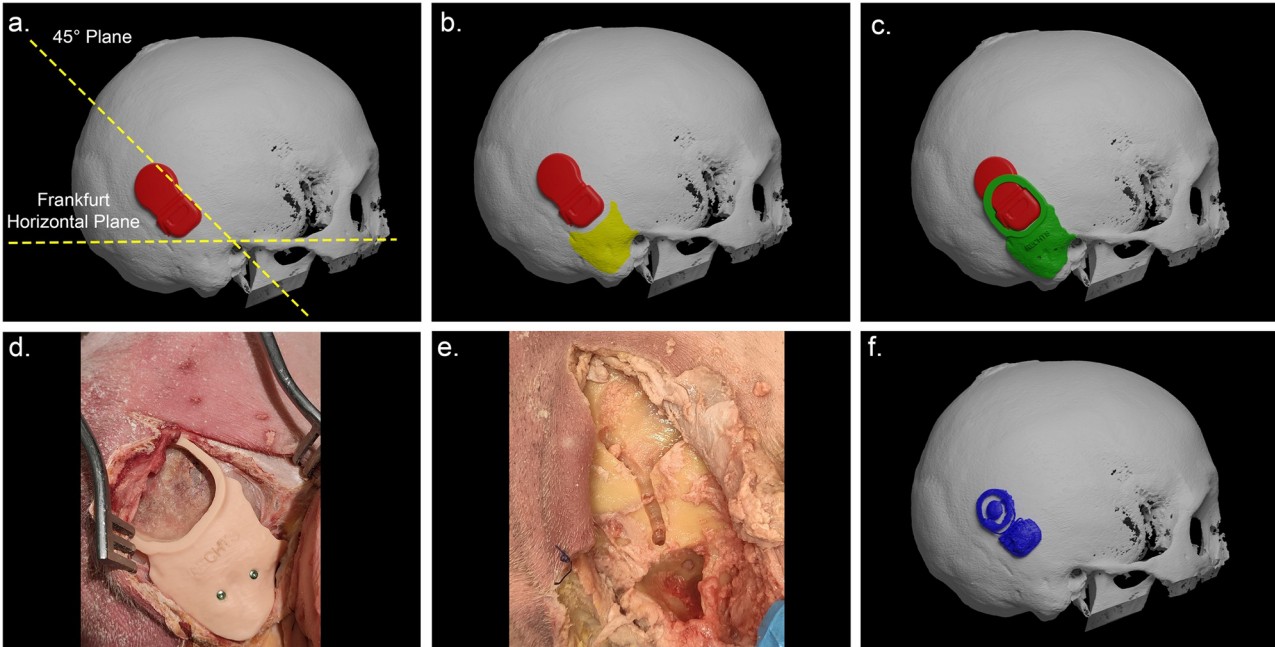

**Fig 1. Planning, guide design and surgical procedure using the 3D-printed guide on a cadaveric head.** (a) The 3D model of the cochlear implant (CI, shown in red) aligned with a 45˚ plane relative to the Frankfurter Horizontal plane, originating from the external auditory canal (EAC). (b) The skull surface of the mastoid bone and the suprameatal crest used as contact areas (marked yellow on the skull). (c) Surgical guide depicted in green. (d) The surgical guide in place on a cadaveric head. (e) Surgical guide removed with a clear view of the drilled cortical recess. (f) Segmented 3D model of the implanted CI based on the postoperative CBCT scan (shown in blue).

place the drilling guide. The guide was secured to the bone with two screws (Fig 1d). A cortical recess was drilled out (Fig 1e), with a bony overhang if bone thickness was adequate. The surgical guide was then removed and the fit of the bony bed was tested by means of a silicone dummy. When the optimal fit was achieved, the cochlear implant was placed in the bony bed and the periosteum was closed, in order to perform the post implantation scan. Each side of a specimen was implanted and scanned sequentially, in order to assess the depth of the bony bed without scattering created by the implant.

## Analysis

After implantation, a CBCT scan was carried out using the same settings as mentioned above. The DICOM images were imported into iPlan and image fusion with the preoperative scan was achieved by first performing manual alignment followed by automated registration based on voxel based matching. Image fusion was visually verified by the researcher. The implanted CI was segmented and exported as a 3D model (Fig 1f). The image fusion step ensured that the pre-implantation 3D models of the CI's and the post-implantation 3D models of the CI's were in the same coordinate system.

In order to compare the accuracy of the CI placement between the specimens we assessed the pre-implantation 3D models to the post-implantation 3D models per case. The 3D models of the CI's were placed in the same coordinate system. This alignment of the CI's between specimens, was achieved by performing the following three steps in 3DMedX (v1.2.11.1, 3D Lab Radboudumc, Nijmegen). First, the 3D models of the CI's were manually placed at the origin of the coordinate system and aligned to the principal axis of this coordinate system, referred to as the centered CI (Fig 2a). Secondly, the 3D model of the planned CI (pre-op) was

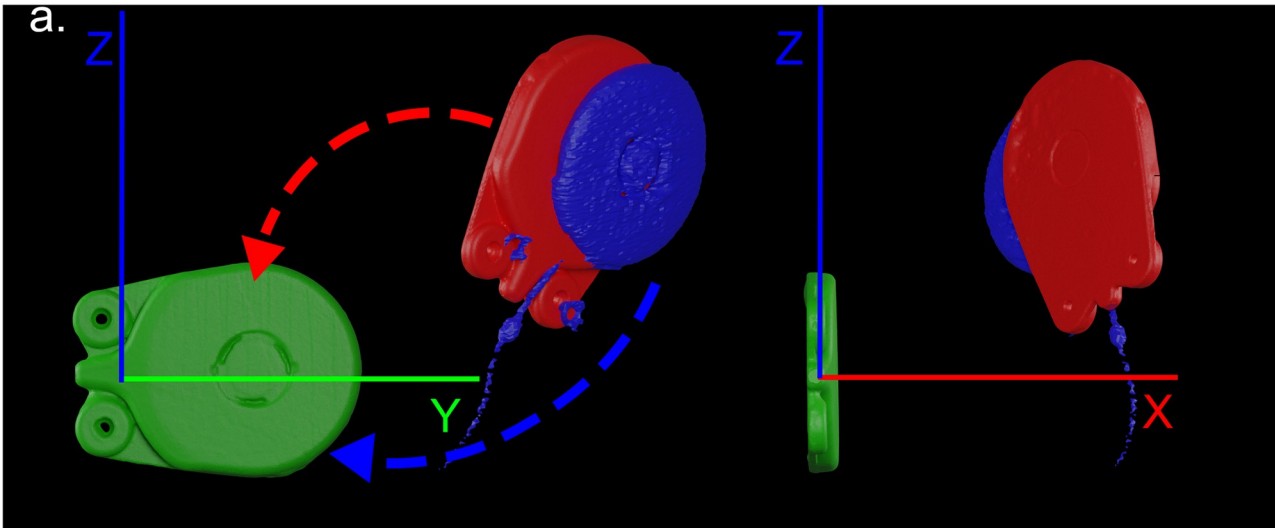

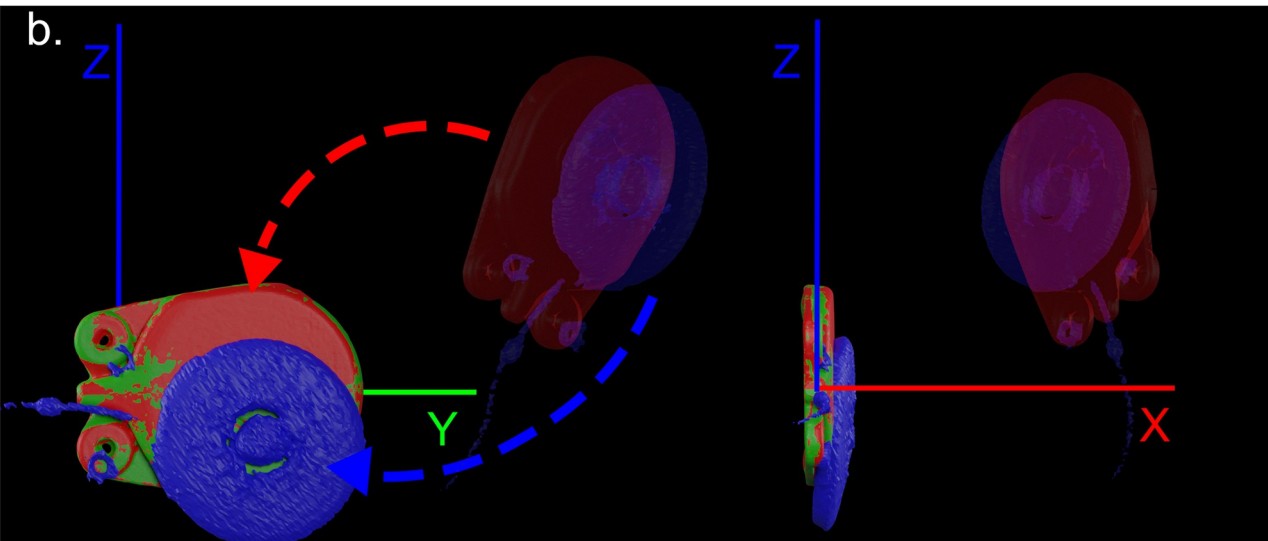

**Fig 2. Analysis steps of the alignment of cochlear implants to eliminate errors due to skull size and planning variability.** The X, Y, and Z axis are marked red, green, and blue, respectively. (a) Depiction of the manually placed cochlear implant (CI) at the origin of the coordinate system (0,0,0) (green color); the 3D model of the planned CI (red color); the 3D model of the implanted CI (blue color). (b) Registration of the planned CI (red color) towards the centered CI model (green color) using rigid surface matching.

registered toward the centered CI of the respective CI model, using rigid surface based matching (Fig 2b). This registration was based on the Iterative Closest Point (ICP) algorithm [9]. An important note is that the 3D models of the centered CI and planned CI were identical, removing the potential of a registration error. Thirdly, the transformation matrix determined by the ICP registration in the previous step was also applied to the 3D model of the implanted CI, extracted from the postoperative CBCT scan (Fig 2b). This placed the implanted CI in the same relative position to the planned for accurate comparison. In order to enable the direct comparison of the left and right implanted CI, the 3D models from the CI's implanted on the left side of the head were mirrored in the sagittal plane before performing the previous three steps.

Finally, the accuracy of the CI placement was determined by performing a second ICP registration from the planned CI, now located at the center of the coordinate system, to the registered 3D model of the implanted CI. The translation (mm) and rotation, expressed as the roll, pitch, and yaw, were derived from the transformation matrix as determined by the second ICP registration. The transformation matrix was converted to the Euler angles using the YXZ sequence. A perfect CI placement would result in a 0 mm translation and 0° rotation along all axis.

Since the combination of a translation and rotation can be difficult to interpret, the accuracy of the CI placement was also expressed as the translation between the center of the magnet of the planned CI and the implanted CI. The center of the magnet only needed to be determined once for each model of CI used in this study, removing a potential observer error of selecting the center of the magnet separately for each cadaver.

## Statistical analysis

Data were analyzed using IBM SPSS Statistics for Windows (version 25.0; IBM Corp., Armonk, NY, USA). Translational and rotational deviations between the planned CI and the implanted CI were analyzed using descriptive statistics. In order to prevent the effect of positive and negative values cancelling each other out, we used the absolute numbers for the statistical analysis. Each ear was analyzed as an individual case. Since we expect the outcome of the study to be not dependent on the characteristic of the specimen, we did not apply adjustment for the correlation between the two ears. This study will be reported according to the guidelines the STROBE statement.

## Results

We implanted and analyzed 9 specimens and 18 ears in total. Specimen 8 was implanted by HT, all other specimens were implanted by LM. Due to outliers, in particular subject 1, 2 and 8, the data were not normally distributed. An overview of the absolute translational and rotational deviations between the planned CI and implanted CI are shown in Table 1. Translational deviation of placement under the 3.0 mm threshold, was achieved in the X- and the Y-axis (median deviation of 1.59 mm with IQR 0.95 and 2.34 mm with IQR 3.84 respectively). Translational deviation was highest in the Z-axis (median deviation of 3.41 mm with IQR 4.55) with also the largest range of deviation. Rotational deviation of placement ranged from 1.53 to 23.73 degrees on the X-axis, 0.10 to 19.55 degrees on the Y-axis and 0.22 to 11.07 degrees on the Z-axis. Specimens number 1 (left side) and 8 (both sides) had Z translational deviations of more than 10 mm (Fig 3a). These cases also had large rotational deviations in the

**Table 1. Absolute translational and rotational deviations between the planned cochlear implant (CI) and the implanted CI calculated with the Iterative Closest Point (ICP) algorithm.**

| | Translational deviations (millimeters) | | | Rotational deviations (degrees) | | |
|---|---|---|---|---|---|---|
| | X-translation | Y-translation | Z-translation | Pitch | Roll | Yaw |
| Median | 1.59 | 2.34 | 3.41 | 5.50 | 4.58 | 3.71 |
| IQR | 0.95 | 3.84 | 4.55 | 8.90 | 6.26 | 4.25 |
| Mean ± SD | 1.65 ± 0.73 | 3.84 ± 3.68 | 4.93 ± 4.95 | 8.02 ± 6.37 | 6.20 ± 5.55 | 4.14 ± 3.20 |
| 95% CI | 1.28–2.01 | 2.01–5.67 | 2.47–7.39 | 4.85–11.18 | 3.43–8.96 | 2.55–5.73 |
| Min | 0.67 | 0.25 | 0.32 | 1.53 | 0.10 | 0.22 |
| Max | 3.48 | 14.33 | 20.30 | 23.73 | 19.55 | 11.07 |

IQR: interquartile range; SD: standard deviation; CI: confidence interval; Pitch: X-rotation; Roll: Y-rotation; Yaw: Z-rotation

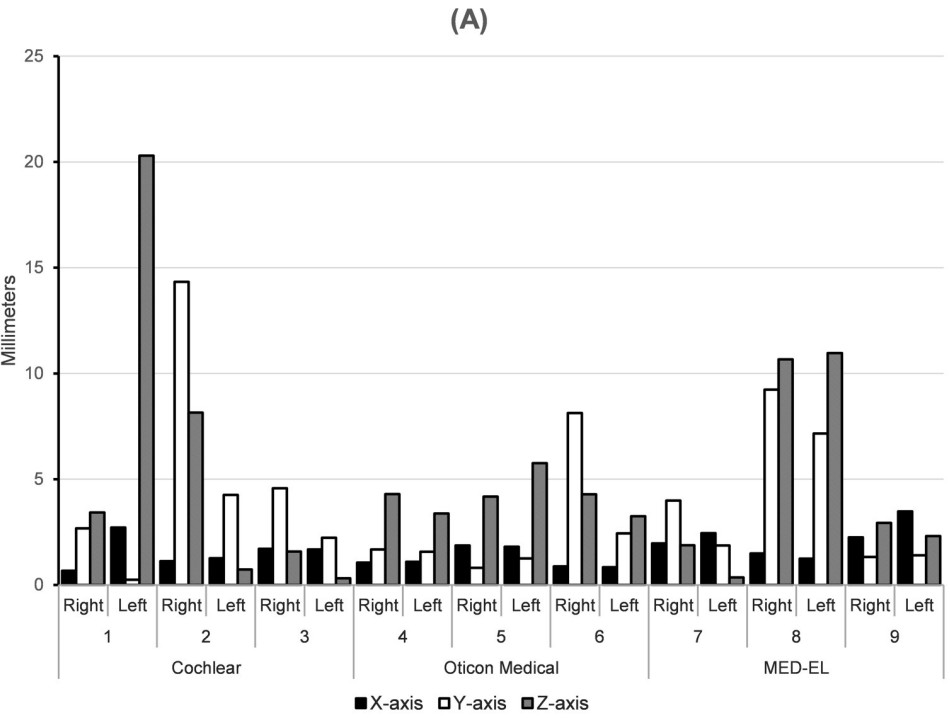

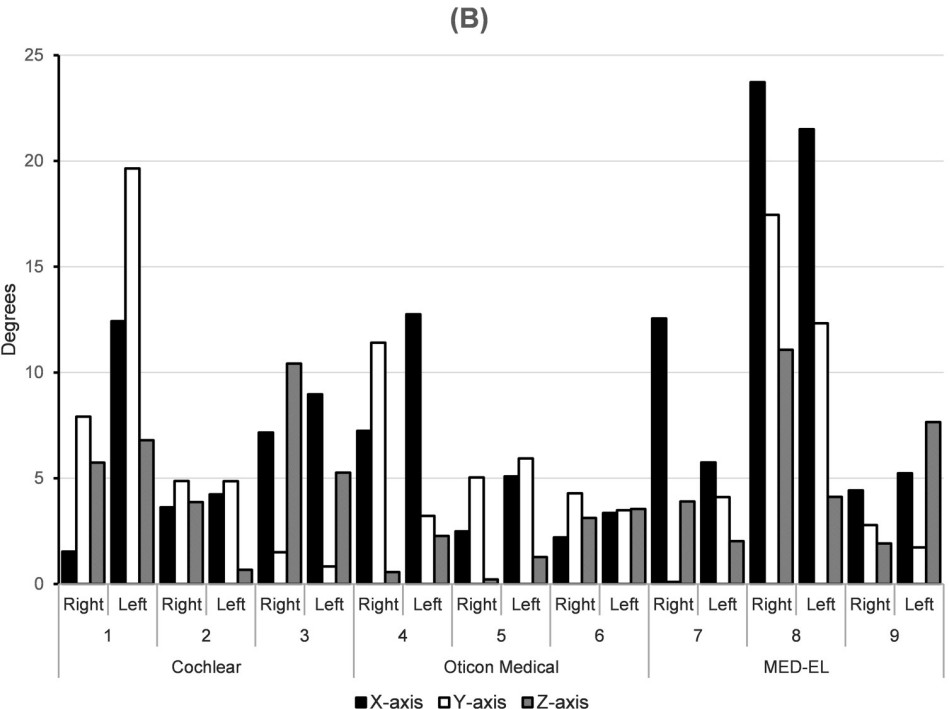

**Fig 3. Translational and rotational deviations (absolute values) per case between the planned CI and implanted CI, expressed in millimeters and degrees.** (a)Translational deviations (absolute values) in millimeters per axis, per case; (b)Rotational deviations (absolute values) in degrees per axis, per case; Horizontal numbers represent the specimens.

**Table 2. Absolute translational deviations (in mm) between the planned cochlear implant (CI) and the implanted CI calculated of the center of the magnet for each CI type with the landmark based analysis.**

|  | X-translation | Y-translation | Z-translation |
|---|---|---|---|
| Median | 1.92 | 2.13 | 4.94 |
| IQR | 2.47 | 3.26 | 5.42 |
| Mean ± SD | 2.46 ± 1.95 | 3.17 ± 3.34 | 7.43 ± 7.92 |
| 95% CI | 1.49–3.43 | 1.51–4.83 | 3.49–11.37 |
| Min | 0.24 | 0.43 | 0.55 |
| Max | 7.20 | 13.85 | 26.05 |

IQR: interquartile range; SD: standard deviation; CI: confidence interval.

X- and Y-axis (Fig 3b). In S1 Table we list the translational and rotational deviations per specimen.

Analysis of the translational deviations between the planned CI and implanted CI calculated for the center of the magnet from each CI, also resulted in median deviations under the 3 mm threshold in the X- and Y-axis respectively (Table 2 for the absolute translational displacement and Fig 4 for the true translation per case). The median translational deviations in the Z-axis was 4.94 mm with IQR 5.42 mm.

## Discussion

In this cadaveric study, we developed a preoperative planning workflow for the positioning and fixation of CI's, and designed a 3D-printed, patient- and CI model-specific surgical guide. The feasibility of using a 3D-printed guide for drilling of the R/S device bony bed was evaluated in conditions as close to reality as possible. To optimize use of the surgical guide screws were added that hold the guide in place, to accommodate the surgeon during the drilling procedure. By staying within 2.5 cm distance from the bony ear canal (which is a stable and reliable landmark visible during preoperative planning on the CBCT), and using the mastoid as well as the external meatus rim and suprameatal crest as landmarks for the surgical guide, more exact positioning on the skull was achieved. The analysis of the planned and implanted CI showed that the median deviations of the X-, and Y-translation were within the predetermined clinically relevant threshold of 3 mm for both landmarks (Tables 1 and 2). Rotational deviations varied between the directions with the Z-rotation having the smallest and X-rotations having the largest deviations (Fig 4).

3D printing is increasingly utilized in otolaryngology in all facets of surgery, from planning to execution [10, 11]. Operative templates in craniofacial and head and neck surgery are mostly used for intraoperative cutting of bony tissues, such as reconstruction of mandibular bony defects [11]. Virtual planning and 3D-printed templates for drilling are less common in otological surgery, although there is increasing interest in applying these techniques in clinical practice. For instance, a method for accurate placement of a bone conduction hearing device has been developed which has shown promising results and has already been used in clinic [12–15]. Another example of surgical templates for drilling, is a study by Vijverberg et al. that used skin-supported guides for bone anchored auricular prostheses [16]. Our study utilized the same principles, applying similar methods in regards to workflow and execution, and faced the same challenges. This study is feasible with any validated software and 3D printers approved for medical use. Furthermore, the preoperative planning and designing of the surgical guide can be realized with different imaging techniques including computed tomography and magnetic resonance imaging [17].

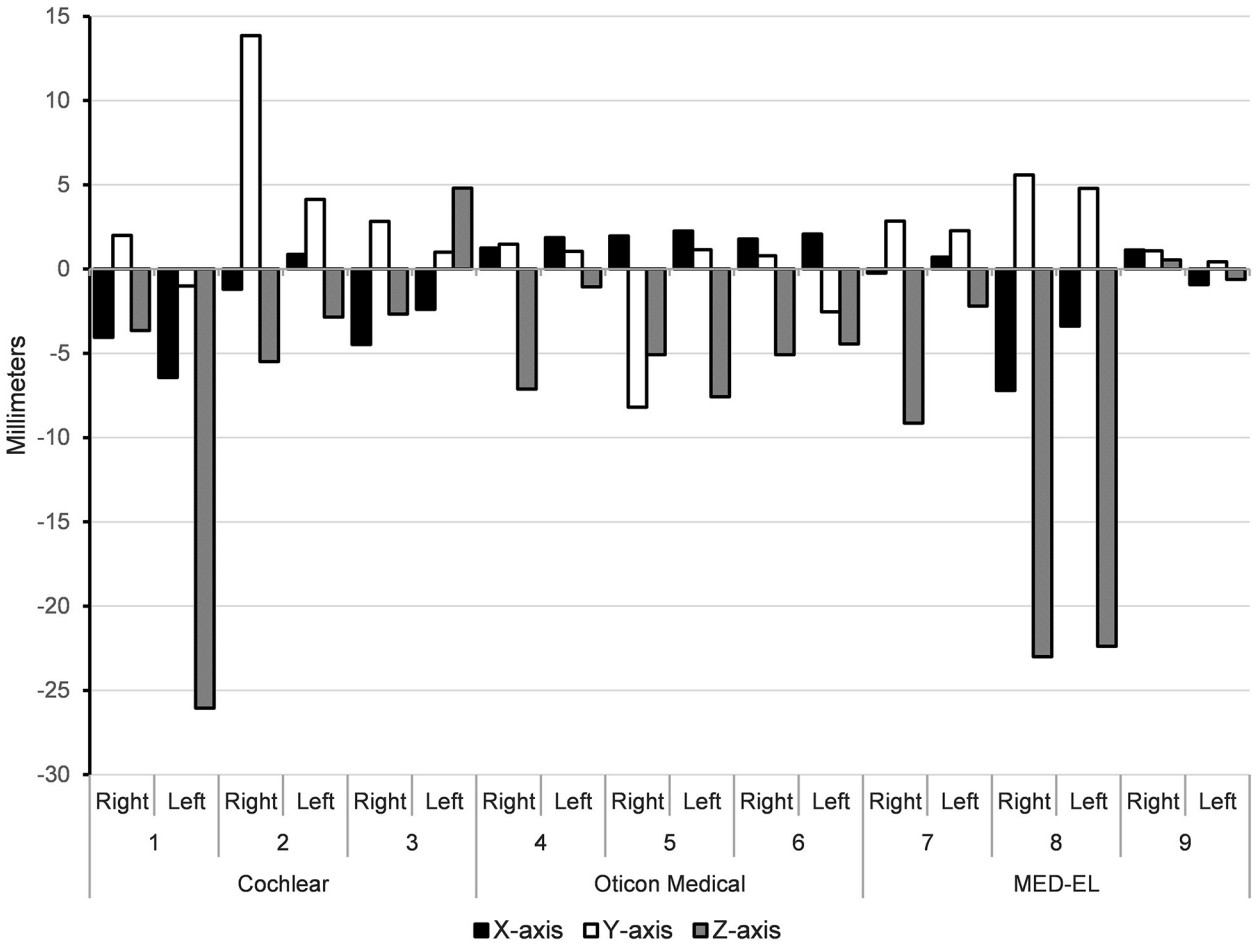

**Fig 4. Translation deviations (true values) of the center of the magnet for each CI type, between the planned CI and the post-op CI per case expressed in millimeters.** Displacement of the center of the magnet between the planned CI and the post-op CI (true values); Horizontal numbers represent the specimens.

This surgical guide is an easy-to-use tool for CI surgeons when drilling a bony bed and optimizes accuracy in regards to positioning on the skull. Moreover, no rough estimates are necessary beforehand when surgically planning the positioning. The template provides the exact location on the skull during surgery. Especially during bilateral cochlear implantation (simultaneous or sequential), it might be a valuable addition to the existing surgical instruments. Symmetrical placement is one of the main aspects visible from outside, regarded as important by these infants' parents, based on our experience. The time invested preoperatively to plan and produce the surgical guide could benefit the surgical procedure by reducing its duration. Furthermore, the process of preoperative planning and production can be automated, making this surgical tool suitable for use in clinic. With the data of this study we cannot conclude if this surgical tool is financially beneficial. This would have to be examined in future clinical studies to weigh the potential reduction of operation time against the costs of production and sterilization.

A challenge we faced during this study was finding the balance between optimizing the surgical guides' accuracy, while also maintaining the low level of invasiveness that is exercised in

clinical practice. A study by Caiti et al. that tested the accuracy of guide positioning on the radius, reported that the accuracy of bone supported surgical guides can vary depending on the location of the bone contact area as well as the size of the surgical guide. They found that extended guides, that is to say guides that covered a larger area of the cortical bone, resulted in a higher placement accuracy [18]. The first designs of our surgical guide had a small contact area and also did not include the mastoid bone. We found that using both the external meatus rim and the mastoid bone as contact areas for the surgical guide gave the best results. These conditions were met by seven cases. The median difference of translation for these cases was under the preset threshold of 3 mm deviation for all translational directions, although the difference with the cases that did not meet these conditions was not statistically significant. The greatest translational improvement using these contact areas was seen in the Z-axis, which was also found by Caiti et al. in their experimental study [18]. Therefore we will use these contact areas when implementing this surgical tool for clinical use. Our results also show a high translational deviation in the Y-axis in these specimens, suggesting a tendency to place the implant more posteriorly. Finally, the translational analysis of the center of the magnet is an easy to interpret analysis of the accuracy which could also be applied in clinic using a flexible tape measure method, validated by our group [19]. Based on the results from this study the largest median deviation would be expected in the Z-axis.

Another point of interest is the apparent learning curve in using the surgical guide. The results of the implantation (only one) executed by HT showed considerable deviation from the planning (Fig 3). This learning curve is to be expected when using a new surgical tool, and this is in line with previous publications of surgical drilling guides [12, 20]. We recommend applying this technique on phantoms such as temporal bones before applying it in vivo.

An important factor that influences accuracy of placement is drilling direction. The surgical tool developed in this study guides the external outline of the bony bed, but it does not guide the direction of the drilling, nor the depth of the bony bed. Due to the fact that the posterior side of the CI (in cases of MedEl, Advanced Bionics and Cochlear, the magnet is situated posteriorly) is not embedded in the skull, the depth of the bony bed is only related to the anterior side of the implant and available cortex thickness. The electrode lead exit also influences the antero-inferior aspect and shape of the bed. Despite these factors influencing the procedure, the translational deviation results of X-translation were satisfactory and evenly distributed between the different implantees, thus we do not expect problems when implementing this method in clinical practice. In this study we used simple guide designs, tested the templates under conditions as close to reality as possible and adhered to a pragmatic accuracy threshold. Satisfactory results were not achieved within the preset limits in all specimens, which is to be expected in a feasibility and pilot study. We identified the potential problems using this tool such as the surgical learning curve as well as the importance of the implant-bone surface contact area, and adapted the design while maintaining a minimally invasive approach. One additional detail is the shape of the retroauricular incision. This should be as minimal invasive as possible (taking into account: scar, pain sensation, esthetics, postoperative morbidity, possible skin related complications) though provide enough space and exposition for adequately drilling a bony well. Therefore in this study a S-shaped "à minima" cut (Lazy S) is applied. It might be discussed whether a C-shaped incision could be opted for (a viable alternative frequently adopted by CI surgeons), however in our experience it does provide insufficient exposure in that region whilst in the same time enough visibility for mastoidectomy and posterior tympanotomy. The optimal skin incision should therefore be included as an objective during future research on this challenging and underestimated topic.

## Conclusion

In this study we developed and tested the first 3D-printed, (patient- and CI model) specific drilling guide. The surgical guide performed well in translational accuracy, and showed more heterogeneity in rotational accuracy. We therefore consider the surgical guide developed in this feasibility study helpful and confirm its potential to increase positioning accuracy in uni- lateral and bilateral cochlear implantations. The next step for the development and evaluation of this surgical guide will be to evaluate the method in clinical practice.

## Supporting information

**S1 Table. Data per cadaver.** All data.
(PDF)

## Acknowledgments

We would like to thank the 3D Face Lab of the Department of Oral and Maxillofacial Surgery at the UMC Utrecht for technical advice and support, and the Department of Anatomy at the UMC Utrecht for supporting this research.

## Author Contributions

**Conceptualization:** Laura M. Markodimitraki, Timen C. ten Harkel, Ronald L. A. W. Bleys, Inge Stegeman, Hans G. X. M. Thomeer.

**Data curation:** Laura M. Markodimitraki, Timen C. ten Harkel.

**Formal analysis:** Timen C. ten Harkel.

**Funding acquisition:** Hans G. X. M. Thomeer.

**Investigation:** Laura M. Markodimitraki.

**Methodology:** Laura M. Markodimitraki, Timen C. ten Harkel, Inge Stegeman, Hans G. X. M. Thomeer.

**Project administration:** Laura M. Markodimitraki.

**Resources:** Ronald L. A. W. Bleys.

**Software:** Timen C. ten Harkel.

**Supervision:** Ronald L. A. W. Bleys, Inge Stegeman, Hans G. X. M. Thomeer.

**Visualization:** Laura M. Markodimitraki, Timen C. ten Harkel.

**Writing – original draft:** Laura M. Markodimitraki, Timen C. ten Harkel.

**Writing – review & editing:** Ronald L. A. W. Bleys, Inge Stegeman, Hans G. X. M. Thomeer.

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
