## [Decision Letter · Decision Letter 0]

4 Feb 2022

PONE-D-21-36271Cochlear implant positioning and fixation using 3D-printed patient specific surgical guides; a cadaveric studyPLOS ONE

Dear Dr. Markodimitraki,

Thank you for submitting your manuscript to PLOS ONE. After careful consideration, we feel that it has merit but does not fully meet PLOS ONE’s publication criteria as it currently stands. Therefore, we invite you to submit a revised version of the manuscript that addresses the points raised during the review process.

Specifically:Although the manuscript is interesting and reads well, both reviewers made comments on the feasibility of such system on daily operations. Furthermore, some of the results could be presented in a more informative manner to reach a broader community of surgeons.

We look forward to receiving your revised manuscript.

Kind regards,

Rafael da Costa Monsanto, M.D.

Academic Editor

PLOS ONE

Journal Requirements:

2. Please include the name and URL of the donation center in your ethics statement in the manuscript Methods and online submission form.

Additional Editor Comments (if provided):

Please address the comments made by the reviewers. Also, please make sure the article suits the journal requirements on competing interest and disclosures

Reviewers' comments:

Reviewer's Responses to Questions

**Comments to the Author**

1. Is the manuscript technically sound, and do the data support the conclusions?

Reviewer #1: Yes

Reviewer #2: Yes

2. Has the statistical analysis been performed appropriately and rigorously? 

Reviewer #1: N/A

Reviewer #2: Yes

3. Have the authors made all data underlying the findings in their manuscript fully available?

Reviewer #1: No

Reviewer #2: Yes

4. Is the manuscript presented in an intelligible fashion and written in standard English?

Reviewer #1: Yes

Reviewer #2: Yes

5. Review Comments to the Author

Reviewer #1: Thank you for the opportunity to review the study titled "Cochlear implant positioning and fixation using 3D-printed patient specific surgical guides; a cadaveric study". Authors developed a 3-D printed surgical guide for cochlear implantation.

Despite the main idea of patient-specific surgical guides and treatment, which I personaly believe is the future of medicine, there are few points I'd like the authors to explain.

1. Why not include Advanced Bionics' devices in the study?

2. In line 113, what is the meaning of STL?

3. In line 136 is said that soft tissue was meticulously removed. During CI surgery, it is preferable not to remove any soft tissue in order to avoid dehiscence of surgical wound.

4. It was not clear if authors found a significant difference between brands.

5. Do authors believe that the 3-D printed surgical guides are financialy suitable for daily activities?

Reviewer #2: Dear authors

The study “Cochlear implant positioning and fixation using 3D-printed patient specific surgical guides; the cadaveric study” aims to create of a surgical guide for a more accurate bone well drilling for the receiver/stimulator of the cochlear implant. I congratulate the authors for the work done. The topic is relevant, having a considerable importance for cochlear implant surgeons.

General Recommendation: Regarding the main theme, I see two questions that could be answered throughout the text. First: what would be the feasibility of carrying out the entire pre-operative planning and 3D printing in other services, using specific software, materials and specific 3d printers for the planning and production of the guide? Second: I believe it would be interesting to discuss in more detail the influence of using those surgical guide on the surgical and planning time.

Intro: The intro writing is good, we can understand the importance of the study, and the challenges that the authors want to solve. I have a specific question on the sentence “The internal part of the cochlear implant, also known as the receiver/stimulator (R/S) device, is designed to reside in close proximity to the pinna, without any interference with the external processor”. What would be the influence of the external processor in the receiver/stimulator? I understand that the external processor cannot be related to the pinna, but the receiver/stimulator will be related with the external processor.

Materials and methods: the authors say that a 45º plane relative to the Frankfurt horizontal plane is originated in the external auditory canal. I suggest the authors to further expand on whether the Frankfurt horizontal plane crosses the external auditory canal, or if it is above or below this. At the same part where the authors say: “the position for the Cochlear and MedEl models was determined so that the anterior part of the implant (receiver/stimulator) would be embedded whilst allowing the posterior part (magnet with coil) to rest on the skull”, something about the OTICON model could be added as well. In the next sentence is important to explain what is the ‘’STL of the cochlear implant’’, there is no mention in the rest of the text about the meaning of STL.

Surgical workflow: the last sentence, ‘’Drilling out of the bony wells of each specimen was performed simultaneously bilaterally, though each side of a specimen was implanted and scanned sequentially, in order to assess the depth of the bony bed without scattering created by the implant.’’ You need to explain how could be possible to drill and implant simultaneously in the same specimen head, maybe you did it bilaterally but not at the same time.

Analysis: it is difficult to understand which comparisons will be made. in my opinion, it would be more important to focus on the idea of comparison between the planned images and the images of the implanted specimen using the surgical guide. This topic point is not explored in depth. The technical details are interesting, but I think there is a need for a more didactic explanation. The help of images is important, but some terms that are not well worked by the author make reading difficult, such as row, pitch and yaw, as well as Euler angles, technical terms that end up making reading difficult and making the passage less interesting

Results: the results are expressed in the text, and in the table with the help of graphics. In the same way as in the topic of analysis, a lot of numerical information is placed, with not sufficient explanation, making the reading and understanding challenging. Even tables and graphs contain a lot of information, but interpretation is extremely difficult. It is difficult to understand which data is important. A more didactic approach would be interesting, better organizing the data, perhaps separating rotational from translational deviations, or some other type of division.

Discussion: The discussion is very well written; it helps a lot in understanding the idea of the study. A few points I would just like to comment:

In the excerpt: ''A challenge we faced during this study was finding the balance between optimizing the surgical guides' accuracy, while also maintaining the low level of invasiveness that is exercised in clinical practice.'' The radiation that patients are exposed in CBCT is one of the invasive points of the IC protocol, especially in young children. In services that only use MRI prior to surgery, would it be possible to carry out planning for the creation of the surgical guide? Question that refers to the initial theme reported by me in the general recommendations on the feasibility of the technique.

When analyzing the contact area of the surgical guide with the temporal bone ''The first designs of our surgical guide had a small contact area and also did not include the mastoid bone. We found that using both the external meatus rim and the mastoid bone as contact areas for the surgical guide gave the best results. These conditions were met by seven cases. '' It suggests that the authors used guides of different sizes in some cases, this should be said in the materials and methods too, I think it would be interesting to report this change in design that occurred throughout the study.

Conclusion: the conclusion of the text is satisfactory.

6. PLOS authors have the option to publish the peer review history of their article (what does this mean?). If published, this will include your full peer review and any attached files.

Reviewer #1: No

Reviewer #2: No

---

## [Author Response · Author response to Decision Letter 0]

2 May 2022

Maybe there has been a confusion, in the last resubmission I revised the data availability statement as requested. However, I received the same feedback as last time. I hope that you can evaluate the current statement.

---

## [Editor Report · Decision Letter 1]

13 Jun 2022

Cochlear implant positioning and fixation using 3D-printed patient specific surgical guides; a cadaveric study

PONE-D-21-36271R1

Dear Dr. Markodimitraki,

We’re pleased to inform you that your manuscript has been judged scientifically suitable for publication and will be formally accepted for publication once it meets all outstanding technical requirements.

Kind regards,

Rafael da Costa Monsanto, M.D.

Academic Editor

PLOS ONE

Additional Editor Comments (optional):

Thank you for addressing the comments made by the reviewers. Congratulations for the excellent work.
---

## [Editor Report · Acceptance letter]

28 Jun 2022

PONE-D-21-36271R1 

Cochlear implant positioning and fixation using 3D-printed patient specific surgical guides; a cadaveric study 

Dear Dr. Markodimitraki:

I'm pleased to inform you that your manuscript has been deemed suitable for publication in PLOS ONE. Congratulations! Your manuscript is now with our production department. 

Kind regards, 

on behalf of

Dr. Rafael da Costa Monsanto 

Academic Editor

PLOS ONE